# USP22 overexpression fails to augment tumor formation in MMTV-ERBB2 mice but loss of function impacts MMTV promoter activity

Xianghong Kuang[1,2], Andrew Salinger[1,2], Fernando Benavides[1,2], William J. Muller[3,4,5], Sharon Y. R. Dent[1,2,6]*, Evangelia Koutelou[1,2]*

1 Department of Epigenetics and Molecular Carcinogenesis, The University of Texas MD Anderson Cancer Center, Houston, TX, United States of America, 2 Center for Cancer Epigenetics, The University of Texas MD Anderson Cancer Center, Houston, TX, United States of America, 3 Rosalind and Morris Goodman Cancer Institute, McGill University, Montreal, Canada, 4 Department of Biochemistry, McGill University, Montreal, Canada, 5 Faculty of Medicine, McGill University, Montreal, Canada, 6 The University of Texas MD Anderson Cancer Center/UTHealth Houston Graduate School of Biomedical Sciences, Houston, TX, United States of America

* sroth@mdanderson.org (SYRD); ekoutelou@mdanderson.org (EK)

**Data Availability Statement:** All relevant data are within the paper and its Supporting Information files.

## Abstract

The Ubiquitin Specific Peptidase 22 (USP22), a component of the Spt-Ada-Gcn5 Acetyltransferase (SAGA) histone modifying complex, is overexpressed in multiple human cancers, but how USP22 impacts tumorigenesis is not clear. We reported previously that *Usp22* loss in mice impacts execution of several signaling pathways driven by growth factor receptors such as erythroblastic oncogene B b2 (ERBB2). To determine whether changes in USP22 expression affects ERBB2-driven tumorigenesis, we introduced conditional overexpression or deletion alleles of *Usp22* into mice bearing the Mouse mammary tumor virus-Neu-Ires-Cre (MMTV-NIC) transgene, which drives both rat ERBB2/NEU expression and Cre recombinase activity from the MMTV promoter resulting in mammary tumor formation. We found that USP22 overexpression in mammary glands did not further enhance primary tumorigenesis in MMTV-NIC female mice, but increased lung metastases were observed. However, deletion of *Usp22* significantly decreased tumor burden and increased survival of MMTV-NIC mice. These effects were associated with markedly decreased levels of both *Erbb2* mRNA and protein, indicating *Usp22* loss impacts MMTV promoter activity. *Usp22* loss had no impact on ERBB2 expression in other settings, including MCF10A cells bearing a Cytomegalovirus (CMV)—driven *ERBB2* transgene or in human epidermal growth factor receptor 2 (HER2)+ human SKBR3 and HCC1953 cells. Decreased activity of the MMTV promoter in MMTV-NIC mice correlated with decreased expression of known regulatory factors, including the glucocorticoid receptor (GR), the progesterone receptor (PR), and the chromatin remodeling factor Brahma-related gene-1 (BRG1). Together our findings indicate that increased expression of USP22 does not augment the activity of an activated ERBB2/NEU transgene but impacts of Usp22 loss on tumorigenesis cannot be assessed in this model due to unexpected effects on MMTV-driven Erbb2/Neu expression.

**Funding:** : "This work was largely supported by the National Institutes of Health grant R01HD094400 to S.Y.R. Dent. This study made use of the Research Animal Support Facility-Smithville (including Laboratory Animal Genetic Services), supported by P30 CA016672 DHHS/NCI Cancer Center Support Grant to MD Anderson Cancer Center and the CPRIT Core Facility Support Grant (RP170628)." The funders had no role in study design, data collection and analysis, decision to publish, or preparation of the manuscript.

**Competing interests:** The authors have declared that no competing interests exist.

## Introduction

Ubiquitin Specific Peptidase 22 (USP22) overexpression occurs in several types of highly aggressive human cancers and is associated with a "death from cancer" gene signature for therapy resistance and stem cell like phenotypes [1–3]. How USP22 impacts cancer development and progression, however, is still not well defined.

USP22 provides the catalytic activity of the deubiquitinase (DUB) module of the Spt-Ada-Gcn5 Acetyltransferase (SAGA) histone modifying complex. SAGA complex is multifunctional, housing both DUB and histone lysine acetyltransferase (KAT) activities, along with factors that interact directly with either sequence specific or general transcription factors to activate genes. Both the KAT and DUB modules of SAGA have been implicated in human cancers [1]. The catalytic subunit of the KAT module, GCN5/KAT2A, is an important co-activator of MYC target genes during normal mouse development and in human cancer cells [4–6]. Deletion or inhibition of *Gcn5* impacts growth and survival of MYC-driven cancers, including lung cancers and lymphoma, in both mouse models and human cells [5, 7]. USP22 functions have also been linked to MYC and other oncogenes, including B lymphoma Mo-MLV insertion region 1 homolog (BMI1), Sirtuin 1 (SIRT1), cyclin B1 and Lysine Demethylase 1A (KDM1A) [8–11]. These findings support a role for USP22 and SAGA in promoting cell proliferation. USP22 impact on proliferation is further indicated by connections between USP22, the androgen receptor and c-MYC in prostate cancer cells and models [12]. Other studies, however, indicate that USP22 also functions to suppress tumor formation in particular contexts, such as colorectal cancer [13]. In all cases, the functions of USP22 in cancer may reflect its activity towards histone substrates and gene regulation or to removal of ubiquitin from non-histone substrates by USP22 that may impact other processes, such as telomere stability or DNA repair [14, 15].

In order to better define USP22 functions in cancer, we created mouse models to determine its functions during normal development. Deletion of *Usp22* leads to embryonic lethality due to defects in placental vasculature [16]. Gene expression changes indicate that these defects are linked to aberrant execution of multiple receptor tyrosine kinase signaling cascades as well as Transforming growth factor beta (TGFβ) signaling that drive propagation of endothelial cells, specification of perivascular cells, and cell migration [16]. These same pathways and processes are often associated with tumorigenesis, so we also created a Lox-Stop-Lox mouse model that overexpresses USP22 in all tissues to determine whether overexpression is sufficient for induction of these pathways. We observed over branching of the mammary glands in USP22 overexpressing mice, along with aberrant up regulation of many of the same signaling pathways that were down regulated in *Usp22* null embryos, including estrogen receptor, extracellular signal-regulated kinase /Mitogen-activated protein kinase (ERK/MAPK), and TGF**β** signaling [16, 17]. However, USP22 overexpression was not sufficient to induce tumors in the mammary gland or in any other tissues [17], suggesting this DUB likely works with specific oncogenes to promote tumorigenesis.

ERBB2 was one of the signaling pathways impacted by *Usp22* loss [16]. Overexpression of ERBB2 (also known as HER2/NEU) is associated with human breast cancers, and mouse models of HER2+ tumors, including MMTV-NIC transgenic mice (expressing an activated oncogenic human *ERBB2*), have provided many insights into how ERBB2 induces tumors and promotes metastases [18–20]. One previous study by others indicated that USP22 regulates the unfolded protein response in HER2+ tumors [21]. Here we combine the MMTV-NIC transgene with either USP22 overexpression or *Usp22* loss to determine whether and how this DUB impacts ERBB2 driven oncogenesis in mice.

## Results

### USP22 overexpression in MMTV-NIC mice

To determine whether Usp22 overexpression might enhance ERBB2-driven formation of mammary tumors, we introduced our previously described *Lox-Stop-Lox (LSL) Usp22* allele [17] into mice carrying the MMTV-NIC transgene (**Fig 1A**). The *LSL Usp22* allele inserted at the *Rosa26* locus allows tissue-specific overexpression (OE) of USP22 upon exposure to Cre recombinase [17]. The USP22 protein expressed from the *OE/LSL* allele (hereafter referred to as *OE*) is fully functional and can fully rescue lethality of *Usp22* null embryos, but it is not sufficient to induce tumors on its own [17]. The MMTV-NIC transgene expresses both the ERBB2/NEU oncoprotein and Cre from the MMTV promoter, assuring Cre expression, and thereby USP22 overexpression, in the same mammary epithelial cells at the same time as ERBB2/NEU overexpression.

We monitored mammary tumor formation in MMTV-NIC mice bearing only the endogenous allele of *Usp22* (*Usp22*$^{+/+}$) or bearing one (*Usp22*$^{OE/+}$) or two (*Usp22*$^{OE/OE}$) copies of the

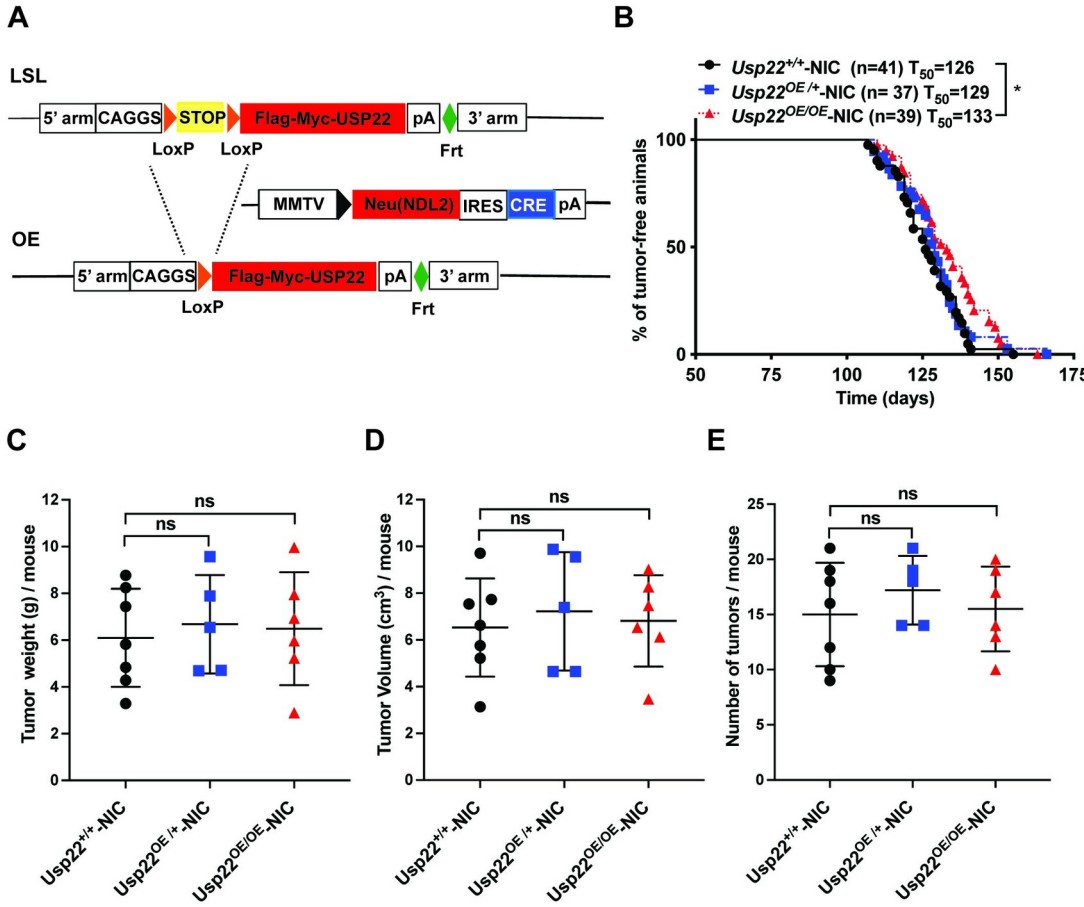

**Fig 1. Overexpression of USP22 does not augment mammary gland tumor formation in MMTV-NIC mice.** (A) Schematic representation of *Rosa26KI-Usp22-LSL* mice that were crossed with MMTV-NIC mice to generate USP22-OE-MMTV-NIC progeny. (B) Kaplan-Meier curve showing tumor-free survival, comparing *Usp22*$^{+/+}$- NIC (n = 41), *Usp22*$^{OE/+}$- NIC (n = 37) and *Usp22*$^{OE/OE}$- NIC (n = 39) mice. *p = 0.012 was determined by Log-rank (Mantel–Cox) test, comparing *Usp22*$^{OE/OE}$- NIC mice with *Usp22*$^{+/+}$-NIC mice. (C-E) Total tumor weight per mouse (C), total tumor volume per mouse (D) and number of tumors per animal (E) were measured in *Usp22*$^{+/+}$- NIC (n = 7), *Usp22*$^{OE/+}$- NIC (n = 5) and *Usp22*$^{OE/OE}$- NIC mice (n = 6). Tumors were harvested 45–55 days after tumor was first palpable (tumor onset). ns: No statistical significance when compared *Usp22*$^{OE/OE}$- NIC mice with *Usp22*$^{+/+}$- NIC mice using two-tailed unpaired t test.

*Usp22 OE* transgene in addition to the endogenous allele (**Fig 1B**). As expected, [22, 23], the MMTV-NIC mice all developed mammary epithelial tumors between ~120–150 days ($T_{50}$, median latency period of 126 days). The kinetics and frequency of tumor formation were not altered by Usp22 overexpression, in either the *Usp22^{OE/+}*; NIC or *Usp22^{OE/OE}*; NIC mice (**Fig 1B**). Usp22 overexpression clearly does not decrease the survival of NIC mice, it consistently leads to a minor increase in survival by 7 days compared to the wild-type mice. No significant changes were observed in tumor weight, tumor volume, or numbers of tumors per mouse upon Usp22 overexpression (**Fig 1C–1E**).

Consistent with these results, no increase in ERBB2, phospho-ERBB2, or ERα protein levels were observed in endpoint tumor lysates isolated from USP22 overexpressing mice, relative to tumor lysates isolated from MMTV-NIC mice (**Fig 2A top**). Downstream cytosolic effectors of the ERBB2 pathway including phosphorylated AKT and phosphorylated ERK also do not show any consistent change upon USP22 overexpression. Interestingly, a trend for increased incidence of lung metastases was observed in both *Usp22^{OE/+}*—NIC or *Usp22^{OE/OE}*—NIC mice (**Fig 2B**), which may be related to decreased expression of E-CADHERIN and increased level of SLUG (Protein snail homolog 2) in *Usp22^{OE/OE}* mice (**Fig 2A bottom**). Expression of other markers of the epithelial-mesenchymal transition, like occludin, vimentin and twist, however, were not significantly changed upon USP22 overexpression. Altogether, these results indicate that increased expression of USP22 does not further enhance the tumorigenicity of the activated ERBB2/NEU oncogene in MMTV-NIC mice.

## *Usp22* deletion impacts tumor formation in MMTV-NIC mice

We next sought to determine whether *Usp22* is required for tumorigenesis in MMTV-NIC mice using the *Usp22^{flox}* (*Usp22^{FL}*) allele, which allows tissue specific deletion of *Usp22* upon exposure to Cre (**Fig 3A**). Loss of one allele of *Usp22* (*Usp22^{FL/+}*) had no impact on the kinetics of tumor formation in MMTV-NIC mice (**Fig 3B**). However, complete deletion of *Usp22* significantly delayed tumor formation ($T_{50}$ 214 days) with some mice remaining tumor free for almost 300 days (**Fig 3B**). Loss of *Usp22* impacted both weight and volume of tumors and substantially reduced the numbers of tumors formed per mouse (**Fig 3C–3E**). These data are consistent with those of Prokakis et al [21], who also observed reduced tumor burden upon loss of *Usp22* using a different MMTV-Neu mouse model.

## Decreased expression of ERBB2 in MMTV-NIC mice

To determine the molecular mechanisms underlying decreased tumor formation, we isolated mammary epithelial cells (MECs) from MMTV-NIC mice with or without the *Usp22^{FL}* allele at an early time point (100 days) (S1 Fig), before any tumors were observed in the MMTV-NIC mice. Immunoblots confirmed partial and complete loss of USP22 protein in the *Usp22^{FL/+}*-NIC and *Usp22^{FL/FL}*-NIC MECs, respectively (**Fig 4A top**). Remarkably, levels of GCN5 and Ataxin 7 Like 3 (ATXN7L3) were also reduced in *Usp22^{FL/FL}*-NIC MECs, indicating potential loss of function of both the histone acetyltransferase and the deubiquitinase modules in the SAGA complex upon *Usp22* deletion (**Fig 4A top**).

Loss of USP22 protein expression correlated with loss of *Usp22* mRNA in *Usp22^{FL/FL}*-NIC MECs (**Fig 4B**). Surprisingly, exogenous *Erbb2* RNA, protein, and phosphorylation levels were also dramatically reduced upon loss of *Usp22* (**Fig 4A bottom** and **4C**). Expression of endogenous mouse ERBB3 was also decreased in *Usp22^{FL/FL}*-NIC MECs, consistent with cross regulation of ERBB2 and ERBB3 stability reported by others [22]. No changes were observed in expression of heat shock protein family members, Heat shock protein 70 (HSP70), Heat shock protein 90 (HSP90), or heat shock protein 90 alpha family class B member 1 (HSP90AB1)

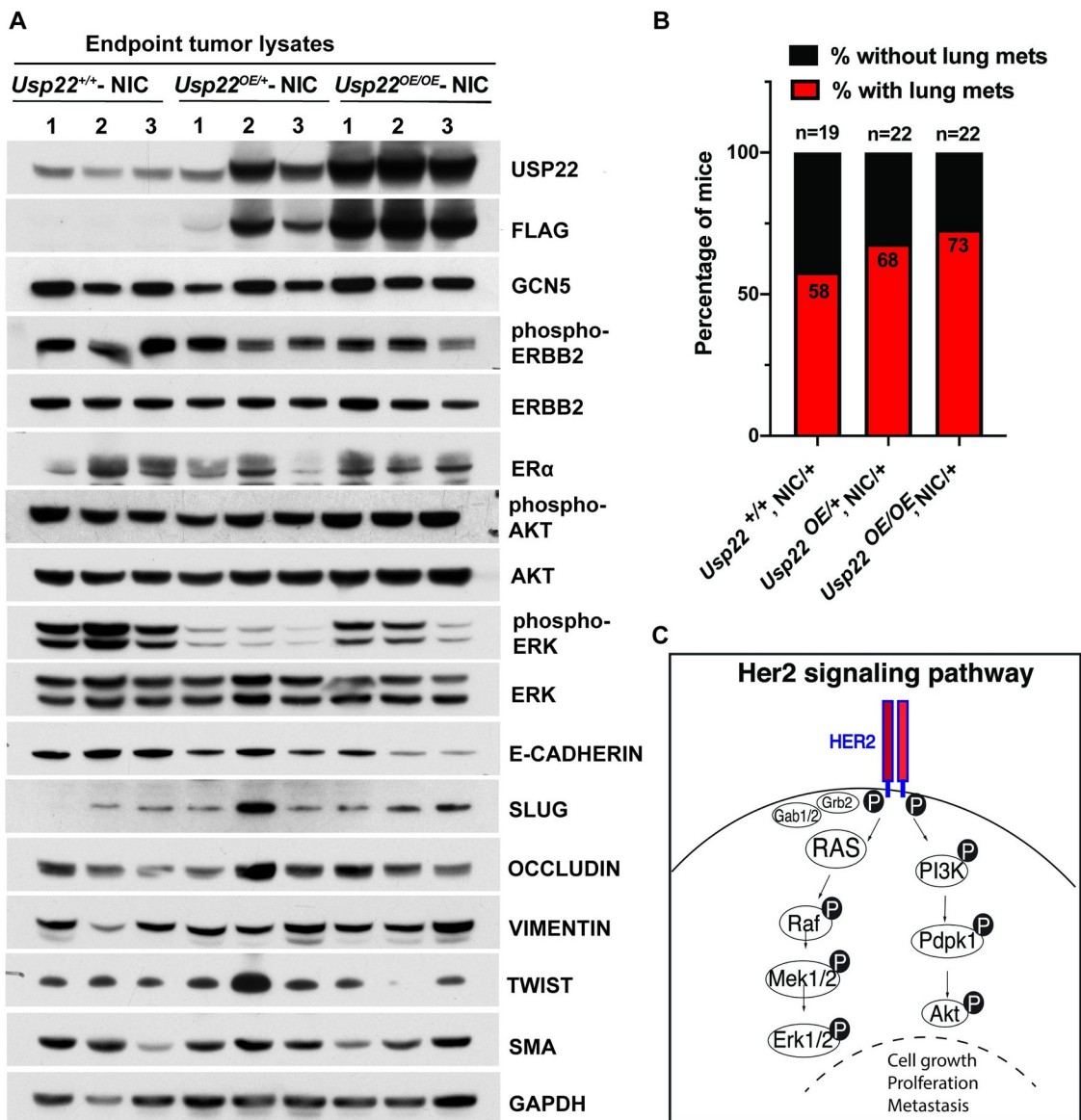

**Fig 2. Expression levels of ERBB2 and ERα proteins and incidence of lung metastases in *Usp22*<sup>OE/OE</sup>- NIC mice.** (A) Immunoblot analyses using lysates from endpoint tumors (n = 3 for each group) showed no difference in the levels of ERBB2 and ERα protein levels upon overexpression of USP22 in *Usp22*<sup>OE/OE</sup>- NIC mice. Levels of phosphorylated ERBB2 protein were also unaffected by USP22 overexpression, as well as the levels of phosphorylated AKT, whereas the levels of phosphorylated ERK show a slight decrease. The levels of EMT markers like E-CADHERIN were decreased, and the levels of SLUG were increased in *Usp22*<sup>OE/OE</sup> NIC/+ mice when compared with *Usp22*<sup>+/+</sup>- NIC mice. EMT markers including OCCLUDIN, VIMENTIN and TWIST were not significantly affected by USP22 overexpression. (B) Incidence of lung metastasis observed in *Usp22*<sup>+/+</sup>-NIC (n = 19), *Usp22*<sup>OE/+</sup>-NIC (n = 22) and *Usp22*<sup>OE/OE</sup>- NIC (n = 22) mice. (C) Graph of Her2 signaling cascade highlighting key cytosolic effectors analyzed in this study.

either in MECs or in end-stage tumors isolated from *Usp22*<sup>FL/FL</sup>—NIC mice (**Fig 4A bottom** and **4D**). These findings are in contrast to those of Prokakis et al, who found no changes in ERBB2/HER2 expression by immunohistochemistry of ERBB2/NEU induced tumors from *Usp22*<sup>FL/FL</sup> mice [21]. This discrepancy may reflect the different time points examined, as our analyses focused on events before tumor formation rather than on the later tumor stages that

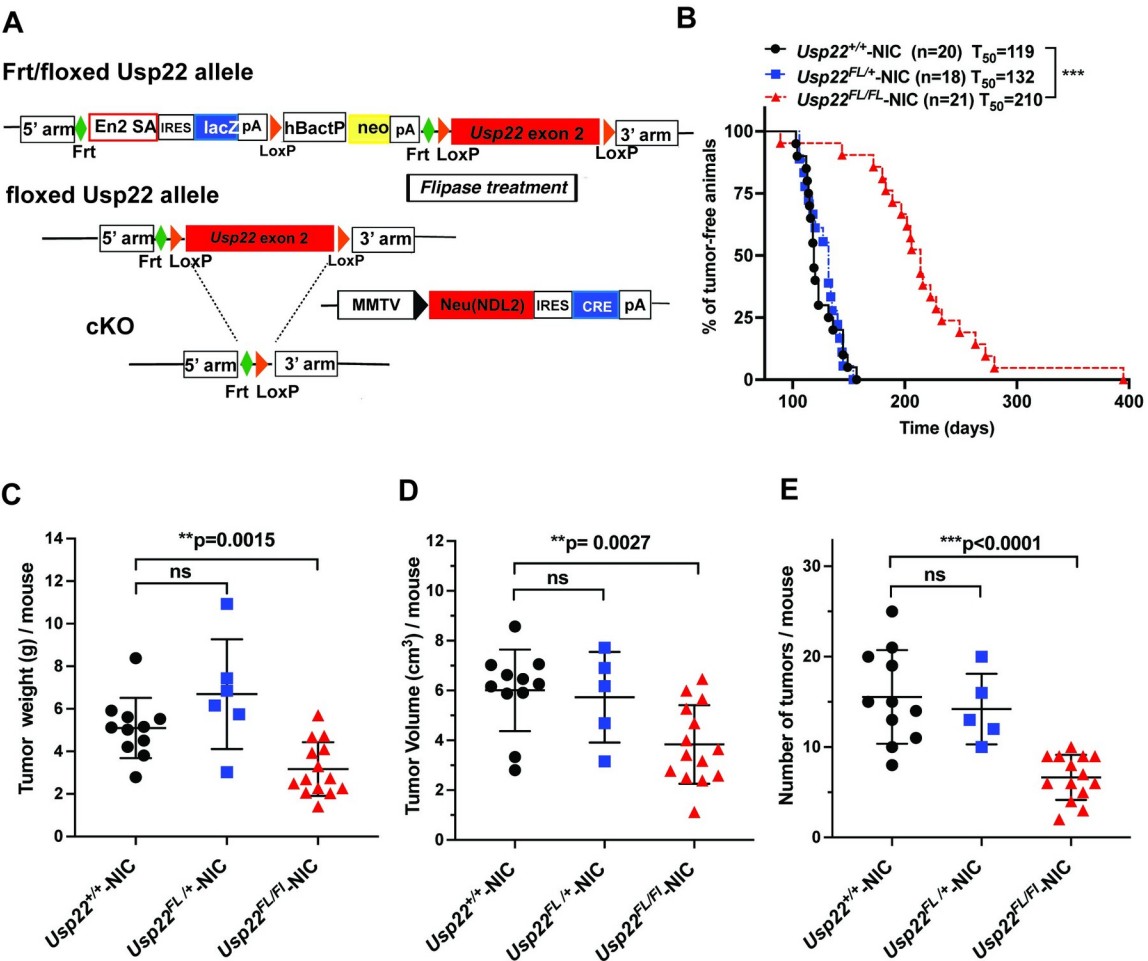

**Fig 3. Delayed tumor formation upon conditional deletion of *Usp22* in mammary gland epithelial cells in MMTV-NIC mice.** (A) Schematic representation *Usp22* floxed mice crossed with MMTV- NIC mice to generate of *Usp22^{FL/FL}*- NIC progeny. (B) Kaplan-Meier curve showing tumor-free survival of animals, comparing *Usp22^{+/+}*- NIC (n = 20), *Usp22^{FL/+}*- NIC (n = 18) and *Usp22^{FL/FL}*- NIC (n = 21) mice. ***p<0.0001 was determined by Log-rank (Mantel–Cox) test where *Usp22^{+/+}*- NIC and *Usp22^{FL/FL}*- NIC were compared. (C-E) Total tumor weight per mouse (C), total tumor volume per mouse (D) and number of tumors per animal (E) measured in *Usp22^{+/+}*- NIC (n = 11; p = 0.0015), *Usp22^{FL/+}*- NIC (n = 5; p = 0.0027) and *Usp22^{FL/FL}*- NIC mice (n = 14; p<0.0001). P values were determined by two-tailed unpaired t test when compared *Usp22^{FL/FL}*—NIC mice with *Usp22^{+/+}*- NIC mice.

did form in these mice. Downstream cytosolic effectors of the ERBB2 pathway such as phosphorylated AKT and phosphorylated ERK are not consistently decreased when the receptors are lost, likely due to the involvement of these effectors in multiple other receptor tyrosine kinase pathways.

The decreased expression of ERBB2/NEU observed in our *Usp22^{FL/FL}*- NIC mice is highly consistent with the decreased tumor burden of these mice. However, these data reveal that the impact of *Usp22* loss on tumorigenesis in this model system, and likely other MMTV driven NEU mouse models, is due to changes in the activity of the MMTV promoter that drives expression of the *Erbb2/Neu* transgene RNA (**Fig 4C**).

### *Usp22* depletion does not affect ERBB2/HER2 expression in human cells

To assess whether *USP22* loss might also impact ERBB2 protein levels independently of *ERBB2* gene transcription changes, we depleted *USP22* in human MCF-10A cells that

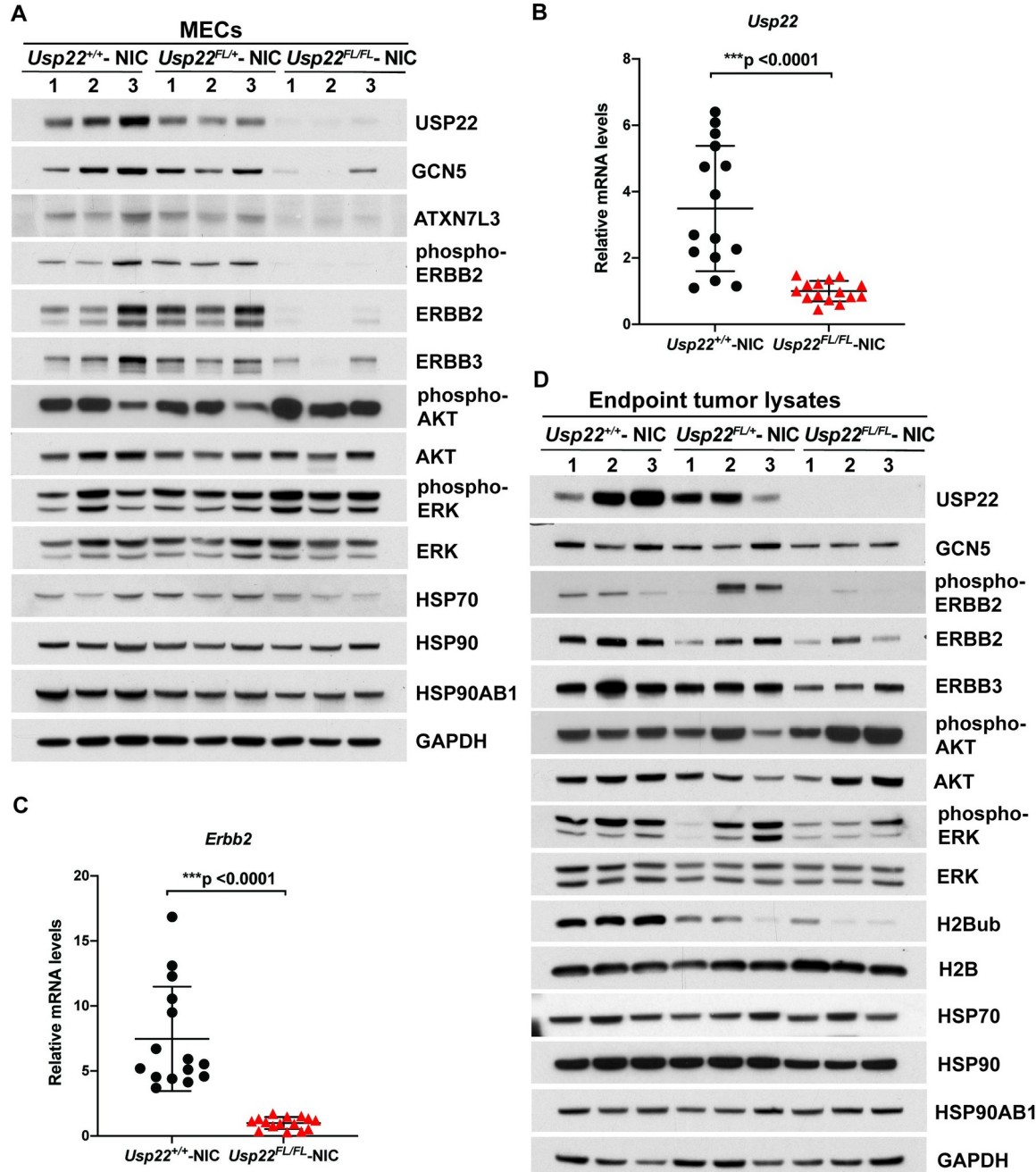

**Fig 4. Decreased expression of SAGA components, ERBB2, and ERBB3 upon deletion of *Usp22* in mammary gland epithelial cells (MECs) collected from 100-day old NIC mice.** (A) Immunoblot analyses showed decreased levels of USP22, GCN5, ATXN7L3, ERBB2, phosphor-ERBB2 and ERBB3 proteins in MECs collected from *Usp22^{FL/FL}- NIC* mice compared to *Usp22^{+/+}—NIC* and *Usp22^{FL/+}—NIC* mice. Despite the sharp decrease in the levels of the receptors, the levels of phosphorylated AKT and phosphorylated ERK show an increase. HSP70, HSP90AB1 and HSP90 protein levels were not affected when analyzed by western blotting in the same lysates. n = 3 for each group. (B, C) qRT-PCR analysis of total RNA collected from *Usp22^{FL/FL}- NIC* MECs confirmed that mRNA levels of *Usp22* and *Erbb2* were drastically decreased compared to *Usp22^{+/+}- NIC*. p<0.0001 for both *Usp22* and *Erbb2*, as determined by two-tailed unpaired t test. n = 5 for each group. Each sample was measured in triplicate. (D) Immunoblot analyses using lysates from endpoint tumors from *Usp22^{FL/FL}- NIC* mice showed that levels of USP22, ERBB2, ERBB3, phosphorylated ERK and H2Bub decreased relative to other genotypes, but levels of HSP70, HSP90, HSP90AB1, GCN5 and phosphorylated AKT did not change. n = 3 for each group.

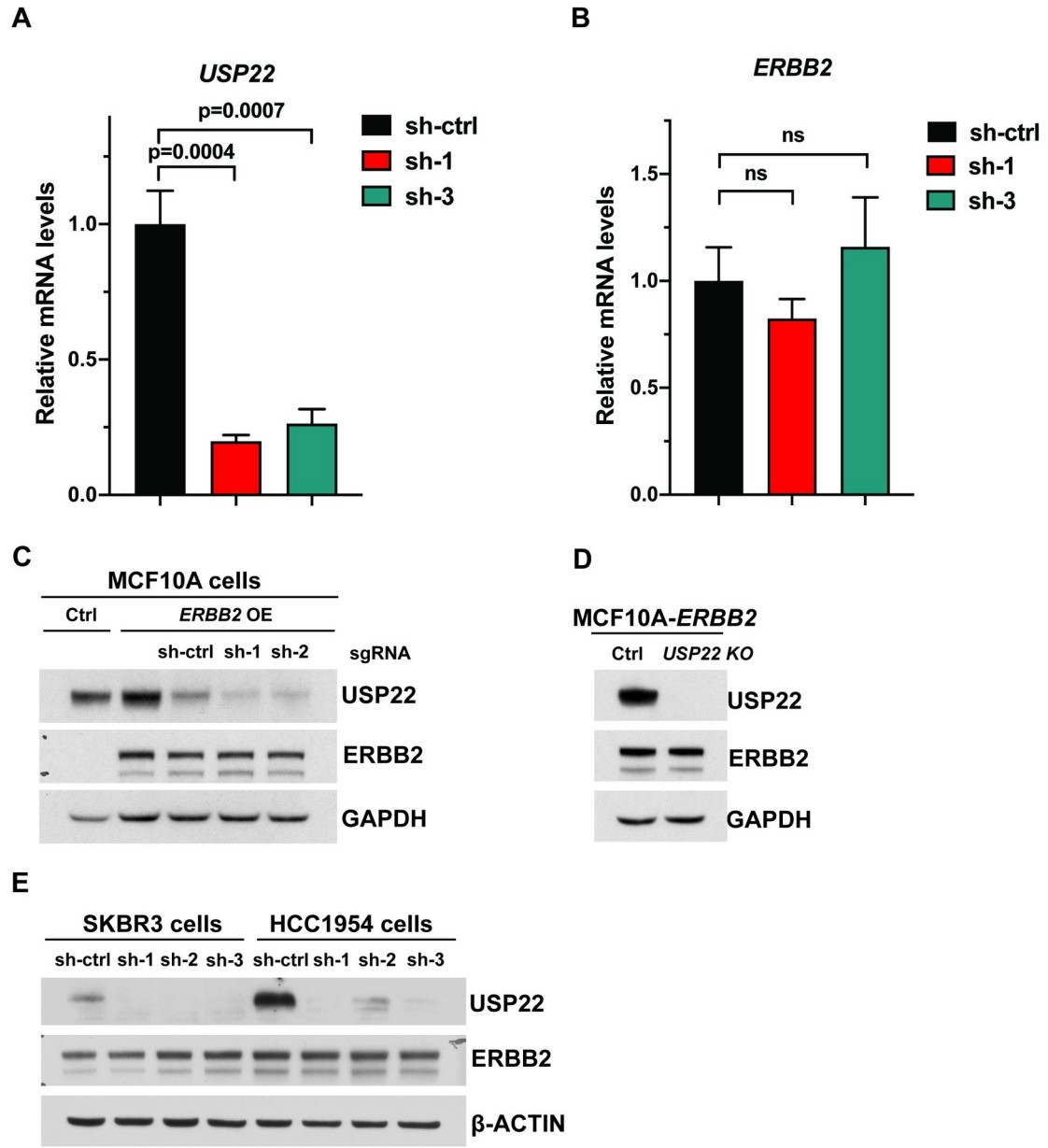

**Fig 5. Depletion or deletion of *USP22* in MCF10A cells that stably overexpress ERBB2 did not affect the levels of ERBB2.** (A) Efficient depletion of *USP22* by lentiviral shRNA knock down in MCF10A-ERBB2 cells as verified by qRT-PCR analysis. (B) qRT-PCR analysis indicates levels of *ERBB2* mRNA did not change upon *USP22* shRNA knock down. ns: No statistical significance. (C) Immunoblot analysis of MCF10A-ERBB2 cell lysates showed that ERBB2 protein levels are not changed upon *USP22* shRNA knock down. (D) Immunoblot analysis of MCF10A-ERBB2 cell lysates indicate that ERBB2 protein levels are not changed upon *USP22* CRISPR/Cas9 knockout. (E). Immunoblot analysis of HER2-positive SKBR3 and HCC1954 cell lysates showed no difference in ERBB2 protein levels upon *USP22* shRNA knock down.

overexpress *ERBB2*. Two independent shRNAs targeting *USP22* significantly depleted both USP22 RNA and protein, with no effects on ERBB2 RNA or protein expression (**Fig 5A–5C**). We further confirmed these results using CRISPR/Cas9 targeting of *USP22*, and again saw no impact on ERBB2 expression upon deletion of *USP22* in the MCF-10A-ERBB2 OE cells (**Fig 5D**). Finally, we examined ERBB2 levels with and without *USP22* depletion in two HER2-positive human breast cancer cell lines, SKBR3 and HCC1954, and again saw no impact on ERBB2

protein levels upon *USP22* loss (**Fig 5E**). These results indicate that *USP22* does not impact ERBB2 expression driven by non-MMTV exogenous promoters or by the endogenous *ERBB2* promoter. However, our results do not rule out effects on downstream factors in the ERBB2 signaling cascade that impact tumorigenesis [24].

## Usp22 loss of function impacts multiple factors required for MMTV promoter activity

MMTV promoter activity is regulated by nuclear hormone receptors including glucocorticoid receptor (GR), progesterone receptor (PR), androgen receptor, as well as by ATP-dependent chromatin remodeling factors [25]. The receptors bind to hormone response elements located in the MMTV promoter and then recruit the Switch/sucrose nonfermentable (SWI/SNF) BRG1 complex and other coactivators to activate the promoter [26–30]. We compared expression of GR, PR, and BRG1 in MECs isolated from 100-day old *Usp22*$^{+/+}$- NIC and *Usp22*$^{FL/FL}$- NIC mice. Loss of Usp22 resulted in decreased levels of *Nr3c*1 transcript and protein (GR), and decreased levels of *PGR* transcript *(*PR*)* (**Fig 6A–6C**). Although *Smarca4* transcript (Brg1) was increased in the absence of *Usp22*, protein levels were decreased (**Fig 6C and 6D**), suggesting USP22 regulates BRG1 at a post-transcriptional level, directly or indirectly. Diminished levels of GR, PR and BRG1 are consistent with decreased MMTV promoter activity and decreased levels of Erbb2 observed in *Usp22*$^{FL/FL}$ -NIC mic*e*.

## Discussion

The role of USP22 overexpression in tumorigenesis and metastasis is an important question, as it is associated with highly aggressive, therapy resistant cancers. Our mouse models clearly established that overexpression of USP22 in mammary epithelial cells on its own is not sufficient to induce tumors [17]. Here, we report that USP22 overexpression does not accelerate or amplify tumorigenesis in MMTV-NIC mice. This lack of effect could indicate a different mode of action for USP22 in tumor formation or progression, or it may reflect the already very high activity and expression of the rat NEU oncogene in these mice. The trend towards increased metastasis to the lung in USP22 OE MMTV-NIC mice may indicate that USP22 facilitates downstream steps in cancer progression, such as the epithelial-mesenchymal transition. Additional studies using a more attenuated oncogene are needed to determine if USP22 augments both tumorigenesis or metastasis.

Increased frequencies of regulator T cells (T-regs) occur in the microenvironment of a variety of malignancies, including breast cancer, and increased numbers of circulating T-regs have been suggested to contribute to the higher metastatic potential of Her-2/neu-positive cells [31]. Others have shown a role for Usp22 in regulating anti-tumor immunity through the modulation of T-regs [32, 33] and future studies using deletion or overexpression of Usp22 in Tregs within a mouse breast cancer model could lead to significant insights to how Usp22 overexpression affects tumorigenesis. In addition, given our previous findings that Usp22 loss impacts placental vascularization [16], it would also be interesting to determine whether overexpression of USP22 in endothelial cells affects tumor progression through enhanced angiogenesis [34].

Our finding that loss of *Usp22* drastically hampers tumor formation in MMTV-NIC mice is consistent with the results of Prokakis et al, who used a similar mouse model wherein ERBB2 expression and Cre were driven by the MMTV promoter, but in separate transgenes. Our data clearly indicate that loss of *Usp22* strongly impacts *Erbb2/Neu* RNA and protein expression in MECs isolated from 100-day old MMTV-NIC mice, before the time of the earliest tumor formation, reflecting decreased activity of the MMTV promoter. Loss of exogenous EBB22/NEU

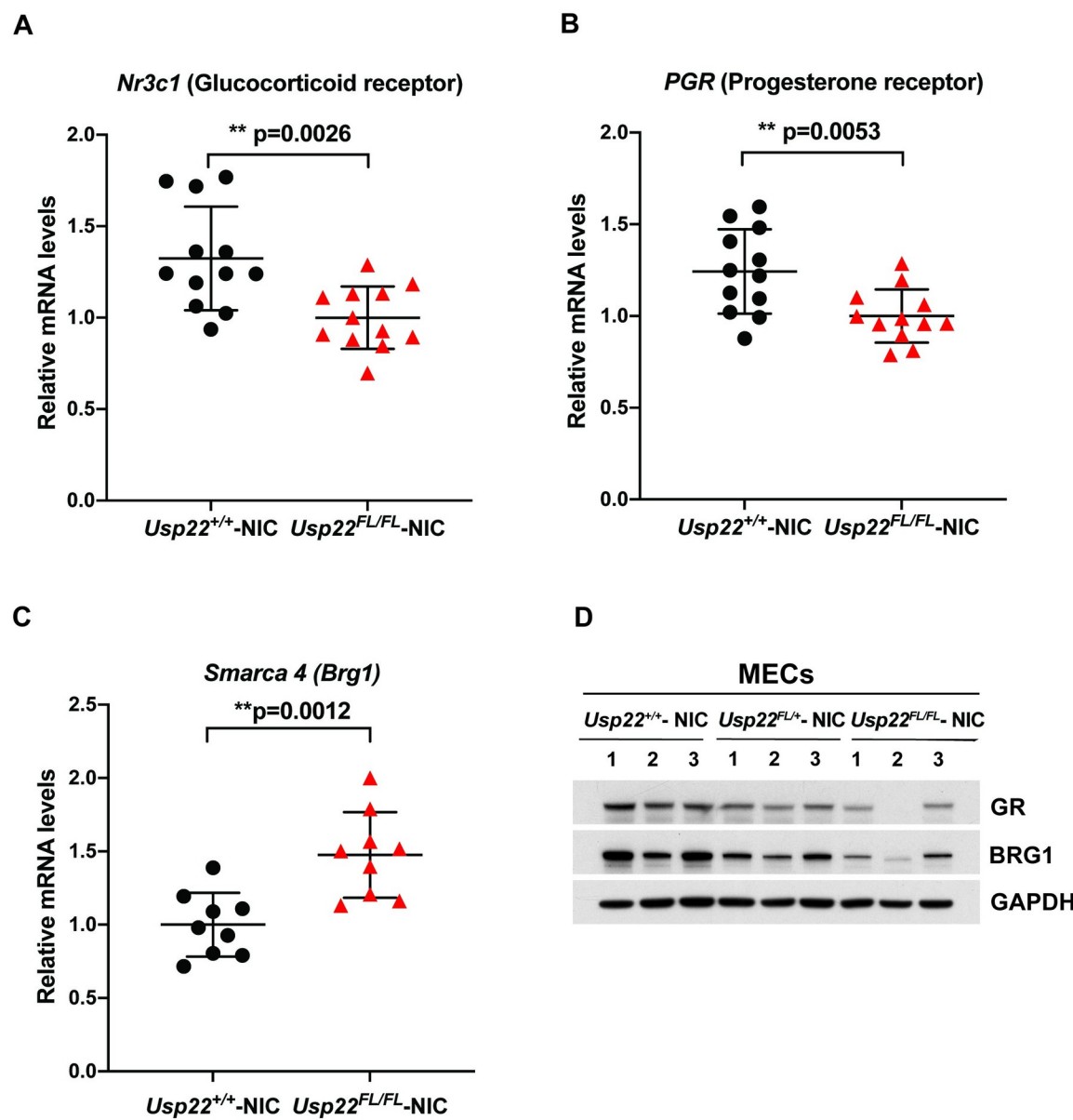

**Fig 6. Expression levels of *Nr3c1* (glucocorticoid receptor), *Pgr* (progesterone receptor) and *Smarca4* (BRG1*) upon loss of *Usp22*.** (A, B, C) qRT-PCR analysis of total RNA collected from *Usp22*$^{FL/FL}$-NIC MECs revealed decreased levels of *Nr3c1* and *Pgr*, but increased levels of *Smarca4 (Brg1)* compared to *Usp22* $^{+/+}$- NIC mice. p = 0.0026 for *Nr3c1*, p = 0.0053 for *PGR* and p = 0.0012 for *Smarca4* were determined by two-tailed unpaired t test. n = 3–4 for each group. (D) Immunoblot analyses indicate decreased levels of GR and BRG1 proteins in MECs collected from *Usp22*$^{FL/FL}$- NIC mice compared to *Usp22* $^{+/+}$- NIC and *Usp22* $^{FL/+}$- NIC mice. n = 3 for each group.

expression also impacted endogenous ERBB3 expression, further reducing tumorigenesis. Reduced levels of ERBB2/NEU protein were also observed in end stage tumors from our MMTV-NIC *Usp22* null mice, although more variability in expression was evident between mice and between tumors. In contrast, Prokakis et al [21] reported no change in ERBB2 levels in their mice, although they only presented measurements of levels of the oncogene in tumors by histology and did not examine pre-tumor cells or tissues. These authors also reported down regulation of the Heat shock protein family A (Hsp70) member 5 (HSPA5), an HSP70 family

member involved in the unfolded protein response, in tumors from *Usp22* null mice, consistent with reports that USP22 mediates deubiquitination of HSPA5 in human prostate cancer cells [15]. Knock down of *USP22* in HCC1954 cells led to loss of HSPA5, indicating USP22 may stabilize HSPA5 to suppress the unfolded protein response. Whether *Usp22* loss in pretumor cells might impact tumor formation through changes in the UPR is not yet clear. Other studies by this group indicated USP22 impacts stability of HSP90AB1 in prostate cancer and breast cancer cells, including HCC1954 cells [35]. We saw no effects on HSP90AB1, HSP90, or HSP70 protein levels in our *Usp22* null MMTV-NIC mice at 100 days or on HSP90AB1 levels in end stage tumors from these mice.

The impact on ERBB2/NEU expression levels that we observe is limited to the MMTV-NIC transgene. Loss of *USP22* had no effect on *ERBB2* RNA or protein expression driven by the CMV promoter in MCF10A cells or on endogenous ERBB2 protein expression in two HER2-positive breast cancer cell lines, SKBR3 and HCC1954. However, these findings do not preclude an impact of *Usp22* loss on downstream components of Erbb2 signaling. Indeed, deletion of *Usp22* in mice or *in vitro*, in induced endothelial cells, led to decreased expression of several common intermediates to receptor tyrosine kinase signaling, including GRB2-Associated Binding Protein 1 (Gab1) and GRB2-Associated Binding Protein 2 (Gab2), and decreased phosphorylation of AKT and ERK1/2 [16]. Additional studies using different, non-MMTV transgene mouse models are needed to determine the impact of *Usp22* loss on these factors in mammary tumors and to define the mechanisms underlying such effects.

Multiple preclinical studies in HER2-positive cells have been performed to understand the molecular mechanism of therapy resistance, but the use of cancer cell lines as the primary model system in those studies does not recapitulate the complexity and heterogeneity of tumorigenesis in organisms. The limitations of conventional cell culture and mouse xenograft studies are well-recognized as an obstacle to the effective translation of preclinical findings into therapy [36]. Mouse models for HER2-driven mammary tumorigenesis have been used extensively and the MMTV-NIC mouse is among the most commonly analyzed [37]. Our work draws attention to caveats in the interpretations of Erbb2 regulation when it is under the control of exogenous promoters like MMTV. Loss of *Usp22* impacts MMTV promoter activity, likely through down regulation of BRG1, PR and GR. These factors themselves impact cancer formation and progression. PR is a well-established breast cancer biomarker in evaluating prognosis and response to treatment. GR plays key roles in breast cancer metastasis and response to treatment [38]. The Cancer Genome Atlas Program (TCGA) analysis reveals high frequency of BRG1 overexpression, but low frequency of BRG1 mutation, in invasive breast cancer [39, 40]. Given the importance of each of these factors in gene regulation and cancer, further delineation of USP22 functions in regulation of BRG1, PR, and GR could provide new insights to its roles in cancer formation and progression. The recent report of novel USP22 specific inhibitors [41] provides strong impetus for fully understanding the functions of this DUB in both normal cells and in cancers.

## Conclusions

Our results demonstrate that USP22 overexpression does not enhance tumorigenesis caused by activated oncogenes like ERBB2/NEU in mice. They also demonstrate that the deletion of *Usp22* in the MMTV-NIC mouse model resulted in a dramatic decrease in tumor burden and an increase in mice survival through the downregulation of the MMTV promoter that drives ERBB2/NEU oncogene expression. Usp22 loss of function, however, did not affect transcriptional or post-transcriptional regulation of ERBB2 in human cells.

## Materials and methods

### Mouse models

Animals were kept in regulated facilities, monitored daily, and all procedures that involved animal handling were performed following the approved Institutional Animal Care and Use Committee (IACUC) protocols (# 00000830-RN03) at the University of Texas MD Anderson Cancer Center (USDA Inspection Report Certificate No. #74-R-0065 and AAALAC Accreditation No #000183).

To generate bigenic mice that specifically overexpress Usp22 in mouse mammary gland epithelium (*Usp22*$^{OE/OE}$; MMTV-NIC), we crossed our lab generated *Rosa26KI-Usp22-LSL* mice (congenic FVB/N background) [17] with MMTV-NIC (MMTV-Neu (NDL2-5)-IRES-Cre, JAX Strain # 032576) mice on pure FVB/N background (generous gift from Dr. William J. Muller) [22]. We identified mice bearing desired genotypes by PCR, using allele-specific primers, 5′ AAAGTCCCTATTGGCGTTACTA 3′ (forward) and 5′ AAAGTCGCTC TGAGTTGTTATC 3′ (reverse) for the knock-in allele with a product size of 388 bp. PCR primers for the wild-type Rosa26 locus had the sequences 5′ GGAGCGGGAGAAATGGATAT 3′ (forward) and 5′ AAAGTCGCTCTGAGTTGTTATC 3′ (reverse), producing a PCR product size of 602 bp. The PCR program ran 35 cycles of 95°C, 5 min, for denaturing; 55°C, 40 sec, for annealing; and 72°C, 1 min, for extension.

To generate mice bearing a *Usp22* floxed allele, we obtained sperm bearing the *Usp22*$^{tm1a(KOMP)Wtsi}$ allele (C57BL/6N) from the Wellcome Trust KOMP consortium (www.komp.org) and performed *in vitro* fertilization. The *Usp22*$^{tm1a(KOMP)Wtsi}$ allele contains two LoxP sites surrounding exon 2 of *Usp22* and a neomycin selection cassette surrounded by two short flippase recognition target (FRT) sites. We crossed mice bearing the *Usp22*$^{tm1a(KOMP)Wtsi}$ allele with FLPeR mice (obtained from the Genetically Engineered Mouse Facility, MD Anderson Cancer Center) expressing a flippase expression construct [42], in order to remove the neomycin cassette. The resulting Usp22$^{FL/+}$ or Usp22$^{FL/FL}$ mice were backcrossed five times onto FVB/NHsd (Envigo, Indianapolis, IN) using marker-assisted (speed congenics) crosses (Laboratory Animal Genetic Services, MD Anderson Cancer Center) [43, 44], and then crossed with MMTV-NIC mice (also in the FVB/N background) to achieve mammary gland epithelial-specific *Usp22* deletion. Genotypes were determined by PCR using primers CSD-*Usp22*-F (forward): 5′ ACTTTGGGAAAGCTCGTATGTGTGC 3′ and CSD-*Usp22*-ttR (reverse): 5′ GATGACATCATCACATTCCCACGCC 3′ yielding a PCR product of 486bp for the wildtype allele and 623bp for the floxed allele.

### Detection of mammary gland tumors and lung metastasis

Standard care, animal health and behavior monitoring were conducted daily by animal facility specialists or research personnel. Mouse mammary gland tumor onset in female NIC/+ mice was determined by palpation twice weekly starting from 10-weeks of age. Tumor size was measured by caliber and when the leading tumor reached 2 cm in the largest dimension mice were euthanized by CO2 inhalation, followed by cervical dislocation per our animal protocol guidelines. The criteria were determined based on published reports from other groups [45–48], our pilot experiments, and animal well-being guidelines approved by the Institutional Animal Care and Use Committee of UT MD Anderson Cancer Center (IACUC animal protocols #00000830-RN03).

All the tumors from each mouse were isolated and measured their weight and volume separately. Single tumor volume was estimated with the formula: volume = (width)$^2$ X length/2 (cm$^3$). Total tumor weight per mouse or total tumor volume per mouse was calculated by combining all the tumor's weight or volume value from each mouse. Lungs were collected, fixed in

10% neutral buffered formalin, and then embedded in paraffin. 5 serial sections at 50 μm intervals from each lung were stained with hematoxylin and eosin and histological evaluated for metastasis lesion.

## Isolation of primary mouse mammary gland epithelial cells (MECs)

The left and right 2nd, 3rd, and 4th mammary glands were dissected from 100-day-old female $Usp22^{FL/FL}$-NIC and wildtype-NIC mice, with lymph nodes removed from the 4th glands. Organoids were prepared from the dissected tissue as described [17, 49] with modifications. Briefly, glands were minced into paste and incubated in 5 mL DME/F-12 medium (HyClone, #SH30023.1) with 2 mg/mL collagenase A (Roche #10103578001), 100 units/mL hyaluronidase (Sigma-Aldrich, H3506) for 1 h at 37°C with 180 rpm rotation. The resulting cell mixture was collected by centrifugation and then washed with DME/F-12 medium and centrifuged twice at 450 g for 10 min to remove the fatty layer. The cells were next incubated in 3 mL DME/F-12 medium with 2 units/mL DNase I (Sigma-Aldrich, D4263) at room temperature for 3 min and centrifuged at 450 g for 10 min. The cell pellets were resuspended in 4 ml Red Blood Cell Lysis buffer (Ammonium Chloride solution, Stemcell Technology, #07800) on ice for 2 min and washed with 6 mL DME/F-12 medium at 450 g for 10 min. Differential centrifugation (pulse centrifugation to 450 g for 10 seconds) was used to separate stroma from organoids. The supernatant from the second spin containing fibroblasts, macrophages, and endothelial cells were discarded. The pellets from 2nd round pulse centrifugations were organoids consisting mainly of mammary gland epithelial cells. The organoids were then digested with 3 mL 0.05% Trypsin/EDTA (HyClone, #SH30236.1) with 2 unit/mL of DNase I for 3 min. An equal volume of DME/F-12 containing 10% fetal bovine serum (FBS, Gibco, #10437–028) was added and the cells were dissociated by pipetting 20 times with a 1000 μL (size) tip. Isolated epithelial cells were then collected by 3 min centrifugation at 600g. The purity of isolated primary MECs was verified by immunoblotting with epithelial (E-CADHERIN CYTOKERATIN 8) and fibroblast (FSP-1) specific markers (S1 **Fig**).

## Cell lines and cell culture

MCF10A cells (normal human mammary epithelial cells) bearing either a vector control or an *ERBB2* overexpression construct (OE) were generously provided by Dr. Dihua Yu [50]. Cells were cultured in DME/F12 media supplemented with 5% horse serum (Thermo Fisher Scientific, 16050122), 20 ng/mL human Epidermal Growth Factor (hEGF) (Sigma, E9644), 100 ng/mL cholera toxin (Sigma, C8052), 0.5 μg/mL hydrocortisone (Sigma, H0888), 10 μg/mL insulin (Sigma, I1882), 50 units/mL penicillin, and 50 μg/mL streptomycin (Hyclone, #SV30010). DMEM media (Hyclone #SH3002201) with 10% FBS was used to culture SKBR3 cells (the American Type Culture Collection (ATCC), #HTB-30), HCC1954 cells (ATCC, #CRL-2338), NMuMG cells (immortalized normal mouse mammary gland epithelial cell line, ATCC, #CRL-1636) and NIH3T3 cells (mouse embryonic fibroblast cells, ATCC, #CRL-1658).

## shRNA knockdown USP22 in SKBR3, HCC1954 and MCF10A-ERBB2 OE cells

HEK293T cells at 70% confluency were co-transfected with lentiviral vectors expressing human *USP22* shRNA. Vectors used included *pGIPZ-GFP-Puro* (see Supplementary Materials) and *psPAX.2* and *pMD2.G* (Addgene). Cell transfections were done in Opti-MEM (Life Technologies, #31985–070) using Lipofectamine 2000 (Life Technologies, Cat. # 11668019) following the manufacturers' transfection reagent manual. Eight hours after the transfection, the medium was changed to DMEM complete medium (DMEM medium with 10% FBS), and 48

hours post transfection, the DMEM medium containing viral particles was collected and filtered through 0.45 μm filters. The virus containing medium with 8 μg/mL polybrene (Sigma-Aldrich, Cat #H9268) was used in 1:1 dilution with the culture medium to infect SKBR3 and HCC1954 cells. 48 hours after infection, the viral medium was changed to DMEM complete medium containing puromycin (EMD Millipore, #540222) at 2 μg/mL for selection of infected cells. Since MCF10A-*ERBB2* cells are already puromycin resistant, transfected cells were sorted on a BD FACSAria Fusion flow cytometer (BD Biosciences) using a 100 μm tip at 70 psi to select for GFP+ cells (comparing transfected cells to un-transfected cells) and were stained with Propidium Iodide to monitor viability. Selected GFP+ cells were collected in a 15mL conical tube with 2mL FBS and then expanded by cell culture in MCF10A culture medium.

### CRISPR-Cas9 USP22 knock out in MCF10A-ERBB2 cells

USP22 was deleted in MCF10A-*ERBB2* OE cells using Synthego Gene Knock out Kit v2, Transfection Optimization Kit and Lipofectamine CRISPRMAX Transfection Reagent (Invitrogen, #CMAX00003) according to Synthego's instructions (www.synthego.com/resources). Briefly, 70% confluent MCF10A-*ERBB2* OE cells were trypsinized, washed and resuspended in OPTI-MEM media. Ribonucleoprotein (RNP) complexes that consist of purified Cas9 nuclease duplexed with chemically modified synthetic multi-guide sgRNA (sgRNA Usp22 KO, from Synthego's Gene Knockout Kit v2) at a ratio of 1:1.3, or a non-sgRNA control in Opti-MEM media were then mixed with lipofectamine CRISPRMAX transfection reagent at a volume ratio of 50:3. The RNP-transfection solution mixture was then mixed with MCF10A-*ERBB2* OE cell suspensions and seeded on cell culture plates. *USP22* knock out cells and non-sgRNA control cells were collected and knock out efficiencies were validated by immunoblot analyses.

### Immunoblotting

Isolated MECs, pelleted cells or tumor samples were lysed in RIPA buffer (150 mM NaCl, 0.1% SDS, 50mM Tris (pH 7.4), 0.5% deoxycholic acid, 1% Nonidet P-40) with protease inhibitor cocktail (Sigma, #P8340) and phosphatase inhibitors PhosSTOP (Roche, #04906837001). The extracts were mixed with Laemmli buffer and boiled at 95˚C for 5 min. The lysates (20–30 μg of total protein) were subjected to electrophoresis on NuPAGE 4–12% Bis-Tris gels (Invitrogen, NP0322BOX), transferred to PVDF membranes (BIO-RAD, #1620177), and probed with primary and secondary antibodies. Antibody information is available in Supplementary Materials. ECL reagents were from Amersham (#RPN2236).

### RNA isolation and qRT-PCR

Total RNA from mouse tissues, cell lines or mammary gland epithelial cells were extracted using RNeasy Plus Mini Kit (Qiagen, #74136). Extracts were then treated with RNase free DNase (Qiagen, #79254). qRT-PCR products were measured using Power SYBR Green RNA-to-Ct 1-step kit (Applied Biosystem, #4389986) and the 7500 Fast Real-Time PCR System (Applied Biosystems). Sequences for gene-specific qRT-PCR primers used are listed in the Supplementary Materials. Expression of *GAPDH* (human) or *Pbgd* (mouse) was used as internal control. Data were normalized by the expression levels of internal control and the relative expression of RNAs was calculated using the comparative Ct method.

### Statistical analysis

All data were represented as mean ±standard error of mean. Calculations for qRT-PCR were performed in Microsoft Excel. The Log-rank (Mantel–Cox) test was used as statistical analysis

for Kaplan-Meier tumor-free survival curves. All other statistical analysis was performed in Prism 9 (GraphPad Prism Software 9.0) using two-tailed unpaired t test. * p value <0.05, ** p< 0.01 and ***p<0.001.

## Supporting information

**S1 Fig. Validation of high enrichment of epithelial cells vs fibroblasts in lysates from mammary epithelial cells (MECs) collected from 100-day old NIC mice.**
(TIF)

**S2 Fig. Raw images of all western blots presented in the upper panel of Fig 2A.**
(TIF)

**S3 Fig. Raw images of all western blots presented in the lower panel of Fig 2A.**
(TIF)

**S4 Fig. Raw images of all western blots presented in Fig 4A.**
(TIF)

**S5 Fig. Raw images of all western blots presented in Fig 4D.**
(TIF)

**S6 Fig. Raw images of all western blots presented in Fig 5.**
(TIF)

**S7 Fig. Raw images of all western blots presented in Fig 6D.**
(TIF)

**S8 Fig. Raw images of all western blots presented in S1 Fig.**
(TIF)

**S1 Table. qRT-PCR primers.**
(XLSX)

**S2 Table. Antibodies.**
(XLSX)

**S3 Table. USP22 lentiviral shRNA vectors.**
(XLSX)

## Acknowledgments

We would like to thank Pam Whitney at the Flow Cytometry and Cell Imaging Core Facility in the Department of Epigenetics and Molecular Carcinogenesis for the support on GFP+ cell sorting experiments. We thank Debra Hollowell for assistance with mouse *in vitro* fertilization. We thank Dr. Lezlee Coghlan, Amanda Martin and animal facility staff for their help in determining tumor size and animal husbandry. We also thank Dr. Dihua Yu (MD Anderson Cancer Center) for providing us MCF10A-vec and MCF10A-*ERBB2* OE cell lines, and Dr. László Tora for providing the anti-ATXN7L3 antibody. We would also like to thank all members of the Dent lab for their advice in experimental procedures.

## Author Contributions

**Conceptualization:** Xianghong Kuang, Sharon Y. R. Dent, Evangelia Koutelou.

**Data curation:** Xianghong Kuang, Evangelia Koutelou.

**Formal analysis:** Xianghong Kuang, Sharon Y. R. Dent, Evangelia Koutelou.

**Funding acquisition:** Sharon Y. R. Dent.

**Investigation:** Xianghong Kuang.

**Methodology:** Xianghong Kuang, Fernando Benavides, William J. Muller.

**Project administration:** Xianghong Kuang, Sharon Y. R. Dent, Evangelia Koutelou.

**Supervision:** Sharon Y. R. Dent, Evangelia Koutelou.

**Validation:** Xianghong Kuang, Andrew Salinger.

**Visualization:** Xianghong Kuang, Evangelia Koutelou.

**Writing – original draft:** Xianghong Kuang, Sharon Y. R. Dent, Evangelia Koutelou.

**Writing – review & editing:** Xianghong Kuang, Sharon Y. R. Dent, Evangelia Koutelou.

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
