## [Decision Letter · Decision Letter 0]

5 Jun 2023

PONE-D-23-12578USP22 overexpression fails to augment tumor formation in MMTV-ERBB2 mice but loss of function impacts MMTV promoter activityPLOS ONE

Dear Dr. Koutelou,

Thank you for submitting your manuscript to PLOS ONE. After careful consideration, we feel that it has merit but does not fully meet PLOS ONE’s publication criteria as it currently stands. Therefore, we invite you to submit a revised version of the manuscript that addresses the points raised during the review process.

We look forward to receiving your revised manuscript.

Kind regards,

Sudhir Kumar Rai, Ph.D

Academic Editor

PLOS ONE

Reviewers' comments:

Reviewer's Responses to Questions

**Comments to the Author**

1. Is the manuscript technically sound, and do the data support the conclusions?

Reviewer #1: Yes

Reviewer #2: Yes

Reviewer #3: Yes

Reviewer #4: Yes

Reviewer #5: Yes

2. Has the statistical analysis been performed appropriately and rigorously? 

Reviewer #1: Yes

Reviewer #2: I Don't Know

Reviewer #3: I Don't Know

Reviewer #4: Yes

Reviewer #5: Yes

3. Have the authors made all data underlying the findings in their manuscript fully available?

Reviewer #1: Yes

Reviewer #2: No

Reviewer #3: Yes

Reviewer #4: Yes

Reviewer #5: Yes

4. Is the manuscript presented in an intelligible fashion and written in standard English?

Reviewer #1: Yes

Reviewer #2: Yes

Reviewer #3: Yes

Reviewer #4: Yes

Reviewer #5: Yes

5. Review Comments to the Author

Reviewer #1: This manuscript reflects work by enthusiasm scientists with a good track record in the field of Usp22 and cancer biology.

Here, the authors identified that modulation Usp22 has different role in MMTV-ERBB2 mediated cancer progression and metastasis.

The present work has been carried out generally with a good experimental standard. The results are presented clearly and the manuscript is well written.

I have some comments and would be great if they address them.

Figure 1: Could they show the level of Usp22 at mRNA or protein level.

Figure 1: They have high number of mice. It would be better if they could add more mice to the analysis of figure 1.E

Figure 3: It would be better if they add blot of figure to show the expression of Usp22 in the three groups.

Figure 3: They have high number of mice. It would be better if they could add more mice to the analysis of Ups22 FL/+_NIC group in figure 3.C-E.

Figure 3.b: Please explain which groups that underwent Kaplan-Meier curve and showed significant difference.

Did they observe any change in the organ weight?

Figure 6: Have they investigated blocking the altered hormones in the context of Usp22 modulation?

It will be valuable if the authors elaborate a little bit more the role of the changed in the hormones in the discussion section.

Reviewer #2: Response to “USP22 overexpression fails to augment tumor formation in MMTV-ERBB2 mice but loss of function impacts MMTV promoter activity” PONE-D-23-12578

Building upon their prior work, the authors describe the effects of Usp22 expression and deletion on ERBB2-driven tumors using mouse and cell line models. Though some of the authors’ results differ from those described previously[1], the authors address this and suggest a possible reason for the discrepancy.

Recommendation

Appropriate for PLOS ONE with minor revision.

Major Concerns

No major concerns.

Minor Concerns

This project appears to have required a substantial amount of experimental work, as evidenced by the authors’ Methods section. However, the authors’ Conclusions section is brief, suggesting no direction for their future work or potential applications of the knowledge generated by their findings. I believe expanding their Conclusions section with such statements would add value for intended readers. Future work could perhaps suggest an attempt to reproduce and improve upon the results of [1] by examining tumors at multiple time points throughout the tumor formation process. Separately, the authors could discuss how their experimental models or findings may contribute to future research investigating Usp22 and its role in the immune landscape of cancer[2,3] or the metastatic progression[4].

Judging by the figures, the statistical analyses appear correct but raw data does not appear to be included in this submission. Specifically, the authors state calculations were performed in 1) Microsoft Excel (pp. 13, line 413) and 2) GraphPad Prism (pp. 13, line 415) but no excel or prism files are included in this submission.

References

[1] Prokakis E, Dyas A, Grün R, Fritzsche S, Bedi U, Kazerouni ZB, Kosinsky RL, Johnsen SA, Wegwitz F. USP22 promotes HER2-driven mammary carcinoma aggressiveness by suppressing the unfolded protein response. Oncogene. 2021 Jun 10;40(23):4004-18.

[2] Guo J, Zhao J, Fu W, Xu Q, Huang D. Immune evasion and drug resistance mediated by USP22 in cancer: Novel targets and mechanisms. Frontiers in Immunology. 2022;13.

[3] Wang Y, Sun Q, Mu N, Sun X, Wang Y, Fan S, Su L, Liu X. The deubiquitinase USP22 regulates PD-L1 degradation in human cancer cells. Cell Communication and Signaling. 2020 Dec;18:1-3.

[4] Zhang K, Yang L, Wang J, Sun T, Guo Y, Nelson R, Tong TR, Pangeni R, Salgia R, Raz DJ. Ubiquitin-specific protease 22 is critical to in vivo angiogenesis, growth and metastasis of non-small cell lung cancer. Cell Communication and Signaling. 2019 Dec;17(1):1-7.

Reviewer #3: This paper seeks to elucidate the role of USP22, which has been shown to play important roles in a variety of human cancers, using breast cancer mouse models. The USP22 Conditional KO model demonstrated the potential of USP22 inhibitors as a therapeutic agent for breast cancer. The constructs of the mouse model are well-established, and the protein expression and phenotype, such as tumorigenesis, have been thoroughly examined. The results were not all as expected, but they were informative for subsequent investigators. With considering the artificial expression of USP22 by the MMTV promoter, the authors discussed results and concluded without over-interpretation.

Given the above, I consider the study as acceptable if the following points for improvement and questions are cleared.

1. Please show full spelling for USP22, SAGA, and many other acronyms. It will help to guess their roles, even for those new to the field.

2. Fig 2A: In the mouse USP22 over expression model, the expression of ERBB2 and phopho-ERBB2 appears to be suppressed. Could this be a dominant-negative effect of USP22? Performing IHC on human breast cancer tissue or qPCR using RNA extracted from the human breast cancer tissue might be able to evaluate the results.

3. Line 221-223: The authors mentioned that “The trend towards increased metastasis to the lung in USP22 OE MMTV-NIC mice may indicate that facilitates downstream steps in cancer progression, such as the epithelial-mesenchymal transition.” Since they have done the Western Blot of Slug, Vimentin, E-cadherin and Twist shown in Fig. 2A, it would be appreciated to have more detailed discussion in relation to the Blot results. If there is no mention in the Discussion, it would be better to send unmentioned results to Supplementary Data.

4. Line 299: The expression “backcrossed several times” is insufficient information. It should be more specific, such as “more than nine times”.

5. Line 356: Was NIH3T3 used in this study?

6. Line 125, 126 and Fig. 1B: The authors stated that “Usp22 overexpression clearly does not decrease the survival of NIC mice”. Was it “clearly decrease”? Consistent with this, could USP22+/+ and USP22OE/OE be opposite each other in Fig. 1B?

7. For ease of comparison, the proteins shown in Fig. 2A and Figs. 4A and 4D should be unified and arranged in the same order throughout three figures. Please include ERBB2/pERBB2, ERK/pERK, and AKT/pAKT.

8. If possible, it would be helpful to illustrate the expected signal and molecule locations based on the Western blot results for better understanding.

Reviewer #4: The authors studied details about the role of USP22 in tumorigenesis. Experimental plan, Data analysis and perper is formated well. The author have explained the study properly and overall it looks great.

Reviewer #5: This study set out to determine whether changes in USP22 expression affects tumorigenesis driven by ERBB2. To achieve this, conditional overexpression and deletion alleles of Usp22 were introduced into mice bearing the MMTV-NIC transgene which drives ERBB2-NEU expression.

Results displayed in Figure 1, clearly demonstrate that the overexpression of USP22 does not further enhance tumorigenesis in MMTV-NIC mice. No significant difference was observed in survival and measurements of tumor weight and volume in USP22-OE-MMTV-NIC progeny.

Results shown in Figure 2 suggest that increased expression of Usp22 may lead to increased lung metastasis. Expression loss of E-cadherin and increased expression of Slug are given as possible reasons for increased metastasis. Other mechanisms may lead to a similar trend in metastasis because e-cadherin remains unchanged in some metastatic tumors. In addition epithelial to mesenchymal transition is not a requirement for metastasis. However, this can be explored in future studies.

Figures 3 definitively shows the increased survival and decreased tumorigenicity of decreased expression of Usp22 using this model. Surprisingly, immunoblot and qt-PCR analyses depicted in Figures 4 and 5 showed that decreased levels and depletion of Usp22 results in decreased expression of ERBB2 in the Usp22FL/FL- NIC model. Decreased expression of Erbb2 was not found in other models such as, MCF10A USP22 CRISPR/Cas9 knockout cells that stably express ERBB2 nor in the SKBR3 and HCC1954 cell lines that express endogenous ERBB2.

It was concluded that loss of Usp22 impacts promotor activity of the MMTV-NIC transgene resulting in decreased expression of Erbb2. This is well supported by results demonstrating loss of expression of nuclear hormone receptors known to regulate MMTV promoter activity (Figure 6). Therefore, MMTV-NIC may not the appropriate model to use to define mechanisms of action of diminshed expression Usp22 on tumorigenesis driven by Erbb2.

It is my recommendation that this manuscript is accepted for publication as written.

6. PLOS authors have the option to publish the peer review history of their article (what does this mean?). If published, this will include your full peer review and any attached files.

Reviewer #1: **Yes: **KAMAL Al-Shami

Reviewer #2: No

Reviewer #3: No

Reviewer #4: No

Reviewer #5: No

---

## [Author Response · Author response to Decision Letter 0]

8 Aug 2023

Reviewer #1: 

This manuscript reflects work by enthusiasm scientists with a good track record in the field of Usp22 and cancer biology.

Here, the authors identified that modulation Usp22 has different role in MMTV-ERBB2 mediated cancer progression and metastasis.

The present work has been carried out generally with a good experimental standard. The results are presented clearly and the manuscript is well written.

I have some comments and would be great if they address them.

We thank the reviewer for carefully and thoroughly assessing our manuscript. We clarify below all the comments that they wanted us to address.

Figure 1: Could they show the level of Usp22 at mRNA or protein level.

Figure 1 focuses on the analysis of the tumor formation and tumor isolation in the mice overexpressing USP22 as described. The levels of Usp22 protein from the isolated tumors from these same mice are presented in Figure 2A. We did not present the RNA levels as the protein blots clearly show USP22 overexpression in the “OE” mice.

Figure 1: They have high number of mice. It would be better if they could add more mice to the analysis of figure 1.E

The number of tumors collected per animal (10-20) was high enough to draw statically sound conclusions from the number of mice presented in Fig.1E. Addition of more mice to the analyses is unlikely to change our conclusions that USP22 overexpression had no significant impact on numbers, weight or volume of tumors in this mouse model.

Figure 3: It would be better if they add blot of figure to show the expression of Usp22 in the three groups.

We do show the levels of Usp22 mRNA and protein from MECs isolated from the mice shown in Figure 3 in Figure 4A, 4B. In Fig. 4D, we show protein levels in isolated endpoint tumors from the three mouse genotype groups.

Figure 3: They have high number of mice. It would be better if they could add more mice to the analysis of Ups22 FL/+_NIC group in figure 3.C-E.

The survival curves clearly show that the wild-type and the heterozygous mice both have reduced survival compared to the homozygous mutant mice. The comparison of tumor weight, tumor volume and tumor number in Figure 3C, D and E includes ample numbers of mice for statistical power, allowing to draw firm conclusions. Moreover, our molecular analysis of MECs by immunoblots from these genotypes shown in Figure 4A confirms no changes in any of the factors we analyzed in the heterozygous mutants, except for the expected ~50% decrease in the amount of USP22, consistent with the survival curve data.

Figure 3b: Please explain which groups that underwent Kaplan-Meier curve and showed significant difference.

We apologize for this omission. We have now clearly marked the 2 animal groups that were compared in Fig.3B (wild type and fl/fl) for the calculation of the p value presented. We have also added the description in the figure legend. 

Did they observe any change in the organ weight?

The focus of our study was the mammary gland, since Cre expression is limited to mammary epithelia in this mouse model. We carefully collected mammary epithelial cells or mammary gland tumors for all presented experiments. The isolation of whole mammary glands is technically very challenging. We did not collect any other organs.

Figure 6: Have they investigated blocking the altered hormones in the context of Usp22 modulation?

It will be valuable if the authors elaborate a little bit more the role of the changed in the hormones in the discussion section.

The mouse model we are analyzing generates conditional deletion of Usp22 in the mammary epithelium where GR, PR and AR are expressed. We thought it is more likely that deregulation of Usp22 levels may have a more prominent effect on the receptors themselves than the hormones they interact with. We have not excluded a potential non-cell autonomous role of Usp22 in the mammary epithelium, however we would need to use a different mouse model to test this hypothesis in depth, in order to avoid interference through the deregulation of the exogenous MMTV promoter in the model used here. 

Reviewer #2: 

Response to “USP22 overexpression fails to augment tumor formation in MMTV-ERBB2 mice but loss of function impacts MMTV promoter activity” PONE-D-23-12578

Building upon their prior work, the authors describe the effects of Usp22 expression and deletion on ERBB2-driven tumors using mouse and cell line models. Though some of the authors’ results differ from those described previously[1], the authors address this and suggest a possible reason for the discrepancy.

Recommendation

Appropriate for PLOS ONE with minor revision.

Major Concerns

No major concerns.

Minor Concerns

This project appears to have required a substantial amount of experimental work, as evidenced by the authors’ Methods section. However, the authors’ Conclusions section is brief, suggesting no direction for their future work or potential applications of the knowledge generated by their findings. I believe expanding their Conclusions section with such statements would add value for intended readers. Future work could perhaps suggest an attempt to reproduce and improve upon the results of [1] by examining tumors at multiple time points throughout the tumor formation process. Separately, the authors could discuss how their experimental models or findings may contribute to future research investigating Usp22 and its role in the immune landscape of cancer[2,3] or the metastatic progression[4].

Judging by the figures, the statistical analyses appear correct but raw data does not appear to be included in this submission. Specifically, the authors state calculations were performed in 1) Microsoft Excel (pp. 13, line 413) and 2) GraphPad Prism (pp. 13, line 415) but no excel or prism files are included in this submission.

References

[1] Prokakis E, Dyas A, Grün R, Fritzsche S, Bedi U, Kazerouni ZB, Kosinsky RL, Johnsen SA, Wegwitz F. USP22 promotes HER2-driven mammary carcinoma aggressiveness by suppressing the unfolded protein response. Oncogene. 2021 Jun 10;40(23):4004-18.

[2] Guo J, Zhao J, Fu W, Xu Q, Huang D. Immune evasion and drug resistance mediated by USP22 in cancer: Novel targets and mechanisms. Frontiers in Immunology. 2022;13.

[3] Wang Y, Sun Q, Mu N, Sun X, Wang Y, Fan S, Su L, Liu X. The deubiquitinase USP22 regulates PD-L1 degradation in human cancer cells. Cell Communication and Signaling. 2020 Dec;18:1-3.

[4] Zhang K, Yang L, Wang J, Sun T, Guo Y, Nelson R, Tong TR, Pangeni R, Salgia R, Raz DJ. Ubiquitin-specific protease 22 is critical to in vivo angiogenesis, growth and metastasis of non-small cell lung cancer. Cell Communication and Signaling. 2019 Dec;17(1):1-7.

We thank the reviewer for their thoughtful comments and suggestions. We have expanded our discussion to include remarks about the potential role of Usp22 in anti-tumor immunity and tumor angiogenesis and included the suggested references in our manuscript. We have included all the raw data used for the figures as supporting information carefully following the requirements and guidelines of the journal.

Reviewer #3: 

This paper seeks to elucidate the role of USP22, which has been shown to play important roles in a variety of human cancers, using breast cancer mouse models. The USP22 Conditional KO model demonstrated the potential of USP22 inhibitors as a therapeutic agent for breast cancer. The constructs of the mouse model are well-established, and the protein expression and phenotype, such as tumorigenesis, have been thoroughly examined. The results were not all as expected, but they were informative for subsequent investigators. With considering the artificial expression of USP22 by the MMTV promoter, the authors discussed results and concluded without over-interpretation.

Given the above, I consider the study as acceptable if the following points for improvement and questions are cleared.

We want to thank the reviewer for carefully and thoroughly assessing our manuscript. We address their detailed comments below.

1. Please show full spelling for USP22, SAGA, and many other acronyms. It will help to guess their roles, even for those new to the field.

We thank you for this suggestion; all the acronyms have now been explained throughout the text. 

2. Fig 2A: In the mouse USP22 over expression model, the expression of ERBB2 and phopho-ERBB2 appears to be suppressed. Could this be a dominant-negative effect of USP22? 

Performing IHC on human breast cancer tissue or qPCR using RNA extracted from the human breast cancer tissue might be able to evaluate the results.

The levels of phosphorylated ERBB2 and total ERBB2 show some variability, so we would not feel confident interpreting the changes as suppression. This variability is likely based on tumor to tumor differences. We are very interested in analyses of human breast cancer tissues in the future, but unfortunately these analyses are not within the scope of the present study.

3. Line 221-223: The authors mentioned that “The trend towards increased metastasis to the lung in USP22 OE MMTV-NIC mice may indicate that facilitates downstream steps in cancer progression, such as the epithelial-mesenchymal transition.” Since they have done the Western Blot of Slug, Vimentin, E-cadherin and Twist shown in Fig. 2A, it would be appreciated to have more detailed discussion in relation to the Blot results. If there is no mention in the Discussion, it would be better to send unmentioned results to Supplementary Data.

Usp22 has been shown to affect the levels of E-cadherin and Snail in pancreatic cancer cell lines, similarly to the results we observe for E-cadherin and Slug in mouse mammary tissues (Ning et al., 2014). Usp22 has also been shown to function as a transcriptional co-activator for ZEB1 in hepatocellular cancer cell lines (Zeng et al.., 2022). No thorough studies have been performed on the role of Usp22 in epithelial-mesenchymal transition and metastasis in mouse cancer models, so we were reluctant to draw conclusions from the trend towards increased metastasis to the lung we observed. Future studies will require a more appropriate metastasis mouse model with the deletion or overexpression of Usp22 in order to gain insight into regulation of the above EMT markers by Usp22. 

4. Line 299: The expression “backcrossed several times” is insufficient information. It should be more specific, such as “more than nine times”.

The mice were backcrossed 5 times using marker assisted (speed congenics) crosses in the laboratory animal genetic services in MD Anderson Cancer Center, supervised by Prof. Benavides. We have added this information to our methods.

5. Line 356: Was NIH3T3 used in this study?

We used NIH3T3 cell line as a positive control for the expression of fibroblast marker (FSP1) shown in supplementary Figure 1.

6. Line 125, 126 and Fig. 1B: The authors stated that “Usp22 overexpression clearly does not decrease the survival of NIC mice”. Was it “clearly decrease”? Consistent with this, could USP22+/+ and USP22OE/OE be opposite each other in Fig. 1B?

Usp22 is very commonly referred in the literature as an oncogene, so we initially hypothesized that the overexpression of Usp22 in the NIC mouse model may accelerate the onset of tumor formation. However, we clearly do not observe an acceleration of tumor onset; rather, we observe a slight delay in the tumor onset (5% difference observed) between wild type and Usp22 OE animals. 

7. For ease of comparison, the proteins shown in Fig. 2A and Figs. 4A and 4D should be unified and arranged in the same order throughout three figures. Please include ERBB2/pERBB2, ERK/pERK, and AKT/pAKT.

Thank you for this suggestion. We have now made all 3 figures consistent, by showing the proteins mentioned above in all of them.

8. If possible, it would be helpful to illustrate the expected signal and molecule locations based on the Western blot results for better understanding.

We have added a diagram of the signaling molecules presented in our study to Fig 2C .

Reviewer #4: 

The authors studied details about the role of USP22 in tumorigenesis. Experimental plan, Data analysis and perper is formated well. The author have explained the study properly and overall it looks great.

We greatly appreciate the reviewer’s comment and their positive assessment of our manuscript.

Reviewer #5: 

This study set out to determine whether changes in USP22 expression affects tumorigenesis driven by ERBB2. To achieve this, conditional overexpression and deletion alleles of Usp22 were introduced into mice bearing the MMTV-NIC transgene which drives ERBB2-NEU expression.

Results displayed in Figure 1, clearly demonstrate that the overexpression of USP22 does not further enhance tumorigenesis in MMTV-NIC mice. No significant difference was observed in survival and measurements of tumor weight and volume in USP22-OE-MMTV-NIC progeny.

Results shown in Figure 2 suggest that increased expression of Usp22 may lead to increased lung metastasis. Expression loss of E-cadherin and increased expression of Slug are given as possible reasons for increased metastasis. Other mechanisms may lead to a similar trend in metastasis because e-cadherin remains unchanged in some metastatic tumors. In addition epithelial to mesenchymal transition is not a requirement for metastasis. However, this can be explored in future studies.

Figures 3 definitively shows the increased survival and decreased tumorigenicity of decreased expression of Usp22 using this model. Surprisingly, immunoblot and qt-PCR analyses depicted in Figures 4 and 5 showed that decreased levels and depletion of Usp22 results in decreased expression of ERBB2 in the Usp22FL/FL- NIC model. Decreased expression of Erbb2 was not found in other models such as, MCF10A USP22 CRISPR/Cas9 knockout cells that stably express ERBB2 nor in the SKBR3 and HCC1954 cell lines that express endogenous ERBB2.

It was concluded that loss of Usp22 impacts promotor activity of the MMTV-NIC transgene resulting in decreased expression of Erbb2. This is well supported by results demonstrating loss of expression of nuclear hormone receptors known to regulate MMTV promoter activity (Figure 6). Therefore, MMTV-NIC may not the appropriate model to use to define mechanisms of action of diminshed expression Usp22 on tumorigenesis driven by Erbb2.

It is my recommendation that this manuscript is accepted for publication as written.

We thank you for the thorough reviewing of our manuscript.

---

## [Editor Report · Decision Letter 1]

16 Aug 2023

USP22 overexpression fails to augment tumor formation in MMTV-ERBB2 mice but loss of function impacts MMTV promoter activity

PONE-D-23-12578R1

Dear Dr.Evangelia Koutelou,

We’re pleased to inform you that your manuscript has been judged scientifically suitable for publication and will be formally accepted for publication once it meets all outstanding technical requirements.

Kind regards,

Sudhir Kumar Rai, Ph.D

Academic Editor

PLOS ONE

---

## [Editor Report · Acceptance letter]

18 Sep 2023

PONE-D-23-12578R1 

USP22 overexpression fails to augment tumor formation in MMTV-ERBB2 mice but loss of function impacts MMTV promoter activity 

Dear Dr. Koutelou:

I'm pleased to inform you that your manuscript has been deemed suitable for publication in PLOS ONE. Congratulations! Your manuscript is now with our production department. 

Kind regards, 

on behalf of

Dr. Sudhir Kumar Rai 

Academic Editor

PLOS ONE